# Double/Debiased Machine Learning for Dynamic Treatment Effects

**Greg Lewis**
Microsoft Research
glewis@microsoft.com

**Vasilis Syrgkanis**
Microsoft Research
vasy@microsoft.com

## Abstract

We consider the estimation of treatment effects in settings when multiple treatments are assigned over time and treatments can have a causal effect on future outcomes. We propose an extension of the double/debiased machine learning framework to estimate the dynamic effects of treatments and apply it to a concrete linear Markovian high-dimensional state space model and to general structural nested mean models. Our method allows the use of arbitrary machine learning methods to control for the high dimensional state, subject to a mean square error guarantee, while still allowing parametric estimation and construction of confidence intervals for the dynamic treatment effect parameters of interest. Our method is based on a sequential regression peeling process, which we show can be equivalently interpreted as a Neyman orthogonal moment estimator. This allows us to show root-n asymptotic normality of the estimated causal effects.

## 1 Introduction

Improving outcomes often requires multiple treatments: patients may need a course of drugs to manage or cure a disease, soil may need multiple additives to improve fertility, companies may need multiple marketing efforts to close the sale. To make data-driven decisions, policy-makers need precise estimates of what will happen when a new policy is pursued. Because of its importance, this topic has been studied by many communities and under multiple regimes and formulations; examples include the field of reinforcement learning [34], longitudinal analysis in biostatistics [2], the dynamic treatment regime in causal inference and adaptive clinical trials [19].

This paper offers a new method for estimating and making inferences about the counterfactual effect of a new treatment policy. Our method is designed to work with observational data, in an environment with multiple treatments (either discrete or continuous), and a high-dimensional state. Valid causal inference is necessary to correctly attribute changes in outcomes to the different treatments applied. But it is more challenging than in a static context, since there are multiple causal pathways from treatments to subsequent outcomes (e.g. directly, or by changing future states, or by affecting intermediate outcomes, or by influencing future treatments).

Our work bridges many distinct literatures. The first is the econometrics literature on semi-parametric inference [22, 33, 21, 1, 7, 6]. We extend this literature, which has typically focused on static treatment regimes, to consider a semi-parametric Markovian model with flexible high-dimensional state and more broadly dynamic treatment regimes. We propose an estimation algorithm that estimates the dynamic effects of interest, from observational data, at parametric root-$n$ rates. We prove asymptotic normality of the estimates, even when the state is high-dimensional and machine learning algorithms are used to control for indirect effects of the state, as opposed to effects stemming from the treatments. Our formulation can be viewed as a dynamic version of Robinson's classic partial linear model in semi-parametric inference [33], where the controls evolve over time and are affected by prior

35th Conference on Neural Information Processing Systems (NeurIPS 2021).

treatments. Our estimation algorithm builds upon and generalizes the recent work of [6, 8] to estimate not only contemporaneous effects, but also dynamic effects over time. In particular, we propose a sequential residualization approach, where the effects at every period are estimated in a Neyman orthogonal manner and then peeled-off from the outcome, so as to define a new "calibrated" outcome, which will be used to estimate the effect of the treatment in the previous period.

In doing this, we build on results in the semi-parametric inference literature in bio-statistics [3, 37, 36] on estimating causal effects in structural nested models (see [39, 29, 16, 4] for recent overviews). In particular, our identification strategy for dynamic treatment effects is a variant of the well-studied $g$-formula and $g$-estimation for structural nested mean models (SNMMs) [27, 30, 28, 32, 31, 29, 20, 38]. The works cited above have provided doubly robust estimators for this setting. One practical challenge faced by these estimation approaches is that the nuisance functions required are hard to estimate when there are many treatments or treatments are continuous.

Our approach addresses these challenges by providing a Neyman orthogonal $g$-estimation algorithm for linear structural nested mean models [28, 29], that allows for both continuous and discrete treatments in each time period. We use a two stage algorithm, where in the first stage a sequence of regression and classification models are fitted and in the second stage a simple linear system of equations is solved. This approach also allows for an easy sample-splitting/cross-fitting approach, which allows the use of arbitrary machine learning approaches in the first stage.

While Neyman orthogonality is a weaker condition than double robustness, it is sufficient to achieve robustness to bias introduced by using machine learning to estimate the nuisance parameters. This approach thus permits machine learning for the nuisance model estimation, which is practically important in high-dimensional state space/control variable settings. We view this as the main contribution of this paper to the biostatistics literature. Other contributions are 1) we provide formal guarantees for the case of a single time-series dataset, as opposed to cross-sectional time-series; and 2) we extend to linear structural nested models with heterogeneous coefficients, thus allowing for infinite dimensional target functions. Using the recent framework of orthogonal statistical learning [12], these can also be estimated via machine learning techniques. The resulting algorithm is an extension of the popular in practice RLearner algorithm [24] for heterogeneous treatment effects, but in the dynamic treatment regime.

Our work is also closely related to the work on doubly robust estimation in longitudinal data analysis and in offline policy evaluation in reinforcement learning [23, 35, 26, 18, 17] from the causal machine learning community. However, one crucial point of departure from all of these works is that we formulate minimal parametric assumptions that allow us to avoid non-parametric rates or high-variance estimation. Typical fully non-parametric approaches in off-policy evaluation in dynamic settings, requires the repeated estimation of inverse propensity weights at each step, which leads to a dependence on quantities that can be very ill-posed in practice, such as the product of the ratios of states under the observational and the target policy. Moreover, these approaches are typically restricted to discrete states and actions. Our goal is to capture settings where the treatment can also be continuous (investment level, price discount level, drug dosage), and the state contains many continuous variables and is potentially high dimensional. Inverse propensity weighting approaches, though more robust in terms of the assumptions that they make on the model, can be quite prohibitive in these settings even in moderately large data sets.

Our work is also somewhat related to the online debiasing literature [11, 10, 42, 14], since we perform inference from adaptively collected data. But our work is different in two main ways. First, the inferential target in this paper is more general, since that literature restricts itself to inference on contemporaneous effects (i.e. effect of treatment on same period outcome) or on policy values in the absence of a dynamic endogenous state (i.e. assuming no dynamic effects through a Markovian state variable). Further, our results extend to the more general structural nested mean model setting. The second important difference is that, in our main setting, we assume multiple independent small chains of samples (each from a separate treated unit). This sidesteps one of the key technical difficulties in the literature above, which is performing inference from a single chain of serially correlated and adaptively collected samples. With multiple short independent chains, the main technical difficulty tackled by these works, namely the potential non-convergence of the covariance matrices that are used in the effect estimate, vanishes. In that respect, that literature is closer to our results on a single adaptive chain (see Section 6). However, for our results in this setting, we make a homoskedastic noise assumption in the treatment choice policy (i.e. that the noise term in the treatment policy enters additively and is drawn i.i.d.), which allows us to easily show convergence of the associated

covariance matrices of our Neyman orthogonal estimator and hence again sidestep a key technical difficulty. It is an interesting avenue for future research if this i.i.d. additive noise assumption can be dropped from our results on a single adaptive chain by utilizing ideas from this literature (e.g. adaptive re-weighted variants of our Neyman orthogonal estimator; as for instance described in [11]).

## 2 Model and Preliminaries

We state our main results in a stylized partially linear state space Markov decision process and in Section 7 we show that our results generalize to more complex Markovian models, known in the biostatistics and causal inference literature as Structural Nested Mean Models (SNMMs). For simplicity of exposition we focus on this simplified version, as it captures all the complexities needed to highlight our main contributions. We consider a sequence $\{X_t, T_t, Y_t\}_{t=1}^m$, where $X_t \in \mathbb{R}^p$ is the state at time $t$, $T_t \in \mathbb{R}^d$ is the action or treatment at time $t$ and $Y_t \in \mathbb{R}$ is an observed outcome of interest at time $t$. We assume that these variables are related via a linear Markovian process:

$$\forall t \in \{1, \ldots, m\}: \qquad \begin{aligned} X_t &= A \cdot T_{t-1} + B \cdot X_{t-1} + \eta_t \\ T_t &= p(T_{t-1}, X_t) + \zeta_t \\ Y_t &= \theta_0' T_t + \mu' X_t + \epsilon_t \end{aligned} \qquad (1)$$

where $\eta_t, \zeta_t$ and $\epsilon_t$ are exogenous mean-zero random shocks, independent of all contemporaneous and lagged treatments and states, that for simplicity we assume are each drawn i.i.d. across time with $T_0 = X_0 = 0$. The function p could be any propensity function.

Our goal is to estimate the effect of a change in the treatment policy on the final outcome $Y_m$. More concretely, suppose that were to make an intervention and set each of the treatments to some sequence of values: $\{\tau_1, \ldots, \tau_m\}$, then what would be the expected difference in the final outcome $Y_m$ as compared to some baseline policy? For simplicity and without loss of generality, we will consider the baseline policy to be setting all treatments to zero. We will denote this expected difference as: $V(\tau_1, \ldots, \tau_m)$. Equivalently, we can express the quantity we are interested in do-calculus: if we denote with $R(\tau) = \mathbb{E}[Y_m \mid do(T_1 = \tau_1, \ldots, T_m = \tau_m)]$, then $V(\tau) = R(\tau) - R(0)$. In Appendix A, we will also analyze the estimation of the effect of adaptive counterfactual treatment policies, where the treatment at each step can be a function of the state.

## 3 Identification via Dynamic Effects

Our first observation is that we can decompose the quantity $V(\tau)$ into the estimation of the *dynamic treatment effects*: if we were to make an intervention and increase the treatment at period $m - \kappa$ by 1 unit, then what is the change $\theta_\kappa$ in the outcome $Y_m$, for $\kappa \in \{0, \ldots, m\}$; assuming that we set all subsequent treatments to zero (or equivalently to some constant value). This quantity is the effect in the final outcome, that does not go through the changes in the subsequent treatments, due to our observational Markovian treatment policy, but only the part of the effect that goes through changes in the state space $X_t$, that is not part of our decision process. This effect can also be expressed in terms of the constants in our Markov process as:

$$\forall \kappa \in \{1, \ldots, m-1\}: \theta_\kappa = \mu' B^{\kappa-1} A$$

**Lemma 1.** *The counterfactual value function $V : \mathbb{R}^{d \cdot m} \to \mathbb{R}$, can be expressed in terms of the dynamic treatment effects as: $V(\tau_1, \ldots, \tau_m) = \sum_{\kappa=0}^{m-1} \theta_\kappa' \tau_{m-\kappa}$*

Thus to estimate the function $V$, it suffices to estimate the dynamic treatment effects: $\theta_0, \ldots, \theta_m$. We first start by showing that the parameters $\theta_0, \ldots, \theta_m$ are identifiable from the observational data. Identification is not immediately obvious. Write the final outcome as a linear function of all the treatments and the initial state: $Y_m = \sum_{\kappa=0}^{m-1} \theta_\kappa' T_{m-\kappa} + \mu' \delta_m + \epsilon_m$. The shock $\delta_m$ contains components that are heavily correlated with the treatments, since the shocks at period $t$ affect the state at period $t + 1$, which in turn affect the observed treatment at period $t + 1$. In other words, if we view the problem as a simultaneous treatment problem, where the final outcome is the outcome, then we essentially have a problem of unmeasured confounding (implicitly because we ignored the intermediate states).

However, we show that these dynamic effects can be identified from the observational data via a sequential peeling process, which as we show in the next section, leads to an estimation strategy that achieves parametric rates. This is one of the main contributions of the paper.

**Theorem 2.** *The dynamic treatment effects satisfy the following set of conditional moment restrictions:* $\forall q \in \{0, \ldots, m\}$

$$\mathbb{E}[\bar{Y}_{m,m-q} - \theta_q T_{m-q} - \mu' B^q X_{m-q} \mid T_{m-q}, X_{m-q}] = 0$$

*where:* $\bar{Y}_{m,m-q} = Y_m - \sum_{\kappa=0}^{q-1} \theta_\kappa T_{m-\kappa}$. *Moreover, if we let* $\tilde{T}_{m-q} := T_{m-q} - \mathbb{E}[T_{m-q} \mid X_{m-q}]$ *and we have that the covariance matrix* $J := \mathbb{E}[\tilde{T}_{m-q}\tilde{T}_{m-q}^\top]$ *is invertible, then these conditional moment restrictions uniquely identify* $\theta_q$.

*Proof.* For any $q \in \{0, \ldots, m-1\}$, by repeatedly unrolling the state $X_m$, $q-1$ times we have that:

$$
\begin{aligned}
Y_m &= \theta_0' T_m + \sum_{\kappa=0}^{q} \mu' B^{\kappa-1} A T_{m-\kappa} + \mu' B^q X_{m-q} + \sum_{\kappa=0}^{q-1} \mu' B^{\kappa-1} \eta_{m-\kappa} + \epsilon_m \\
&= \psi_m' T_m + \sum_{\kappa=0}^{q} \psi_{m-\kappa}' T_{m-\kappa} + \mu' B^q X_{m-q} + \sum_{\kappa=0}^{q} \mu' B^{\kappa-1} \eta_{m-\kappa} + \epsilon_m \\
&= \psi_{m-q}' T_{m-q} + \sum_{j=m-q+1}^{m} \psi_j' T_j + \mu' B^q X_{m-q} + \sum_{\kappa=0}^{q} \mu' B^{\kappa-1} \eta_{m-\kappa} + \epsilon_m
\end{aligned}
$$

Thus by re-arranging the equation, we have:

$$\bar{Y}_{m,m-q} = \psi_{m-q}' T_{m-q} + \mu' B^q X_{m-q} + \sum_{\kappa=0}^{q} \mu' B^{\kappa-1} \eta_{m-\kappa} + \epsilon_m$$

Since for all $t \geq m - q$, $\eta_t, \epsilon_t$ are subsequent random shocks to the variables $T_{m-q}, X_{m-q}$, we have that they are mean-zero conditional on $T_{m-q}, X_{m-q}$. Thus:

$$\mathbb{E}[\bar{Y}_{m,m-q} - \psi_{m-q}' T_{m-q} - \mu' B^q X_{m-q} \mid T_{m-q}, X_{m-q}] = 0$$

Thus we have concluded that for any $t \in \{1, \ldots, m\}$:

$$\mathbb{E}[\bar{Y}_{m,t} - \psi_t' T_t - \mu' B^{m-q} X_t \mid T_t, X_t] = 0$$

which concludes the proof of the first part of the theorem.

For the second part, consider any $q$ and let $t = m - q$. Note that by taking an expectation of the above condition for with respect to $T_t$ conditional on $X_t$, we get that it implies

$$\mathbb{E}[\bar{Y}_{m,m-q} \mid X_{m-q}] = \psi_{m-q}' T_{m-q} - \mu' B^q X_{m-q} \mid T_{m-q}, X_{m-q}]$$

$\square$

## 4 Dynamic DML Estimation

We now address the estimation problem. We assume that we are given access to $n$ i.i.d. samples from the Markovian process, i.e. we are given $n$ independent time-series, and we denote sample $i$, with $\{X_t^i, T_t^i, Y_t^i\}$. Our goal is to develop an estimator of the function $V$ or equivalently of the parameter vector $\theta = (\theta_0, \ldots, \theta_m)$. We will consider the case of a high-dimensional state space, i.e. $p \gg n$, but low dimensional treatment space and a low dimensional number of periods $m$, i.e. $d, m \ll n$ is a constant independent of $n$. We want to estimate the parameters $\theta$ at $\sqrt{n}$-rates and in a way that our estimator is asymptotically normal, so that we can construct asymptotically valid confidence intervals around our dynamic treatment effects and our estimate of the function $V$. The latter is a non-trivial task due to the high-dimensionality of the state space. For instance, the latter would be statistically impossible if we were to take the direct route of estimating the whole Markov process (i.e. the high-dimensional quantities $A, B, \mu$): if these quantities have a number of non-zero coefficients that grows with $n$ at any polynomial rate, then known results on sparse linear regression, preclude their estimation at root-n rates (see e.g. [40]). However, we are not really interested in these low-level parameters of the dynamic process, but solely on the low dimensional parameter vector $\theta$. We will treat this problem as a semi-parametric inference problem and develop a Neyman orthogonal estimator for the parameter vector [22, 33, 1, 7, 6].

In particular, we consider a sequential version of the double machine learning algorithm proposed in [6]. In the case of a single time-period, i.e. $m = 0$, then [6], recommends the following estimator for $\theta_0$: using half of your data, fit a model $\hat{q}_0(X_0)$ of $\mathbb{E}[Y_0 \mid X_0]$, i.e. that predicts the outcome $Y_0$ from the controls $X_0$ and a model $\hat{p}_0(X_0)$ for $\mathbb{E}[T_0 \mid X_0]$. Then estimate $\theta_0$ on the other half of the data, based on the estimating equation:

$$m(\theta; \hat{p}_0, \hat{q}_0) = \mathbb{E}\left[ (\tilde{Y}_0 - \theta_0 \tilde{T}_0) \tilde{T}_0 \right] = 0$$

where $\tilde{Y}_0 = Y_0 - \hat{q}_0(X_0)$ and $\tilde{T}_0 = T_0 - \hat{p}_0(X_0)$ are the residual outcome and treatment.

We propose a sequential version of this process that we call *Dynamic DML* for *sequential double/debiased machine learning*. Intuitively our algorithm proceeds as follows:

1. We can construct an estimate $\hat{\theta}_0$ of $\theta_0$ in a robust (Neyman orthogonal) manner, by applying the approach of [6] on the final step of the process, i.e. on time step $T_m$; this will estimate all the contemporaneous effects of the treatments,

2. Subsequently we can remove the effect of the observed final step treatment from the observed final step outcome, i.e. by re-defining the random variable $Y_{m,m-1}^i = Y_m^i - \hat{\theta}_0 T_m^i$; doing this we have removed any effects on $Y_m^i$, caused by the final treatment $T_m^i$.

3. We can then estimate the one-step dynamic effect $\theta_1$, by performing the residual-on-residual estimation approach with target outcome the "calibrated" outcome $Y_{m,m-1}^i$, treatment $T_{m-1}^i$ and controls $X_{m-1}^i$. Theorem 2 tells us that the required *conditional exogeneity* moment required to apply the residualization process is valid for these random variables. We can continue in a similar manner, by removing the estimated effect of $T_{m-1}$ from $Y_{m,m-1}^i$ and repeating the above process.

We provide a formal statement of the Dynamic DML process in Algorithm 1, which also describes more formally the sample splitting and cross-fitting approach that we follow in order to estimate the nuisance models $p$ and $q$ required for calculating the estimated residuals.

---
**Algorithm 1** Dynamic DML
---
Randomly split the $n$ samples in $S, S'$
**for** each $\kappa \in \{0, \ldots, m\}$ **do**
    Regress $Y_m$ on $X_{m-\kappa}$ using $S$ to learn estimate $\hat{q}_\kappa$ of model $q_\kappa(x) = \mathbb{E}[Y_m \mid X_{m-\kappa} = x]$ and calculate residuals $\tilde{Y}_{m,m-\kappa}^i = Y_m^i - \hat{q}_\kappa(X_{m-\kappa}^i)$ on other half; vice versa use $S'$ to learn model and evaluate on $S$.
    **for** each $\tau \in \{0, \ldots, \kappa\}$ **do**
        Regress $T_{m-\tau}$ on $X_{m-\kappa}$ using $S$ to learn estimate $\hat{p}_{\tau,\kappa}$ of model $p_{\tau,\kappa}(x) = \mathbb{E}[T_{m-\tau} \mid X_{m-\kappa} = x]$ on the first half and calculate residuals $\tilde{T}_{m-\tau,m-\kappa}^i = T_{m-\tau}^i - p_{\tau,\kappa}(X_{m-\kappa}^i)$ on other half, and vice versa.
    **end for**
**end for**
Using all the data $S \cup S'$
**for** $\kappa = 0$ **to** m **do**
    Regress $\bar{Y}_{m,\kappa} = \tilde{Y}_{m,m-\kappa} - \sum_{\tau < \kappa} \hat{\theta}_\tau' \tilde{T}_{m-\tau,m-\kappa}$ on $\tilde{T}_{m-\kappa,m-\kappa}$, i.e. find a solution $\hat{\theta}_\kappa$ to:

$$\frac{1}{n} \sum_{i=1}^{n} \left( \bar{Y}_{m,\kappa}^i - \theta_\kappa \tilde{T}_{m-\kappa,m-\kappa}^i \right) \tilde{T}_{m-\kappa,m-\kappa}^i = 0$$

**end for**
---

# 5 Estimation Rates and Normality

Our main theorems are to show that subject to the first stage models of the conditional expectations achieving a small (but relatively slow) estimation error, then the recovered parameters are root-n-consistent and asymptotically normal. Our asymptotic normality proof relies on showing that one can

re-interpret our *Dynamic DML* estimator as a $Z$-estimator[1] based on a set of moments that satisfy the property of Neyman orthogonality[2].

Let $p, q$ denote the vector of all nuisance functions and $p^*, q^*$ their true values. Moreover, let $\theta$ denote the vector of dynamic effect parameters, with $\theta_0$ its true value. We provide both finite sample rates mean squared error rates and asymptotic normality of our estimates (proofs in Appendix B.3 and B.4). Moreover, in the appendix we discuss how the required guarantees for the nuisance functions can be easily satisfied using estimation algorithms such as the Lasso and under sparsity conditions.

**Theorem 3** (Finite Sample). *Suppose that $\mathbb{E}[\zeta_t \zeta_t'] \succeq 2\lambda I$ and that each coordinate $\hat{h}$ of each nuisance model $\{q_\kappa, p_{\tau,\kappa}\}_{\tau,\kappa}$ satisfy that w.p. $1 - \delta$:*

$$\|\hat{h} - h^*\|_2 = \sqrt{\mathbb{E}_X[(\hat{h}(X) - h(X))^2]} \leq \epsilon_{n,\delta}$$

*where expectation is with respect to the corresponding input of each model. Moreover, suppose that:[3]*

$$\left\| \sum_{\tau=0}^{m} |\mathbb{E}\left[ \tilde{T}^*_{m-\tau, m-\kappa} \zeta_t' \right]| \right\|_\infty \leq c_m$$

*Then w.p. $1 - 2\delta$:*

$$\max_{t \in [m]} \|\hat{\theta}_t - \theta_{0,t}\|_2 \leq O\left( d\, C_m \left( \sqrt{\frac{\log(d\, m/\delta)}{n}} + \epsilon_{n,\delta}^2 \right) \right)$$

*where $C_m := \max\left\{1, \left(\frac{c_m}{2\lambda - O(m\, d\, \epsilon_{n,\delta})}\right)^{m+1}\right\}$. If each coordinate $\hat{h}$ of each nuisance model satisfies: $\mathbb{E}_{\hat{h}}[\|\hat{h} - h^*\|_2^4]^{(1/4)} \leq \epsilon_n$, then:*

$$\sqrt{\mathbb{E}\left[\max_{t \in [m]} \|\hat{\theta}_t - \theta_{0,t}\|_2^2\right]} \leq O\left( d\, C_m \left( \sqrt{\frac{\log(d\, m)}{n}} + \epsilon_n^2 \right) \right).$$

**Theorem 4** (Asymptotic Normality). *Suppose that the nuisance models $\{q_\kappa, p_{\tau,\kappa}\}_{\tau,\kappa}$ satisfy that:*

$$\forall \kappa, \tau \leq \kappa : \|q_\kappa - q_\kappa^*\|_2, \|p_{\tau,\kappa} - p_{\tau,\kappa}^*\|_2 \leq o(n^{-1/4})$$

*Moreover, suppose that all random variables are bounded and that $\mathbb{E}[\zeta_t \zeta_t'] \succeq \lambda I$. Then:*

$$\sqrt{n}(\hat{\theta} - \theta_0) \to N(0, V)$$

*where $V = M^{-1}\Sigma(M^{-1})'$ and $M$ is a $m \times m$ block lower triangular matrix consisting of blocks of size $d \times d$, whose block entries are of the form:*

$$\forall \tau \leq \kappa : M_{\kappa,\tau} = \mathbb{E}[(T_{m-\tau} - \mathbb{E}[T_{m-\tau} \mid X_{m-\kappa}]) \zeta_{m-\kappa}']$$

*and $\Sigma$ is a $m \times m$ block diagonal matrix whose diagonal block entries take the form: $\Sigma_{tt} = \mathbb{E}[\epsilon_{m-t}^2 \zeta_{m-t} \zeta_{m-t}']$.*

**Concrete Rates for Lasso Nuisance Estimates.** Suppose that the observational policy $p$ is also linear, i.e. $p(X) = AX$. Then all the models $q_\kappa$ and $p_{\tau,\kappa}$ are high-dimensional linear functions of their input arguments, i.e. $q_\kappa(x) = \phi' x$ and $p_{\tau,\kappa,j}(x) = \pi'_{\tau,\kappa,j} x$. If these linear functions satisfy a sparsity constraint then under standard regularity assumptions we can guarantee if we use the Lasso regression to estimate each of these functions that w.p. $1 - \delta$, the estimation error of all nuisance models is $O\left(s\sqrt{\frac{\log(p/\delta)}{n}}\right)$, where $s$ is an upper bound on the number of non-zero coefficients. One sufficient regularity condition is that the expected co-variance matrix of every period's state has full rank, i.e. $\mathbb{E}[X_t X_t'] \succeq \lambda I$. Thus the requirements of the main theorems of this section would be satisfied as long as the sparsity grows as $s = o(n^{1/4})$. These conditions are for instance satisfied if $s$ coordinates of the high-dimensional state have any effect on the final outcome (i.e. are outcome-relevant) and if $s$ coordinates of the high-dimensional state enter the observational policy.

---

[1] A $Z$-estimator is a solution to an empirical analogue of a vector of moment equations $\mathbb{E}[m(W; \theta, \nu)] = 0$, where $W$ are all random variables, $m$ is a vector valued function and $\theta \in \mathbb{R}^d$ is a vector parameter of interest and $\nu \in \mathcal{V}$ is a potentially infinite dimensional nuisance parameter. The true parameters satisfy $\mathbb{E}[m(W; \theta_0, \nu_0)] = 0$.

[2] A vector of moments satisfies Neyman orthogonality if $\forall \nu \in \mathcal{V}: \frac{\partial}{\partial t}\mathbb{E}[m(W; \theta_0, \nu_0 + t(\nu - \nu_0))]|_{t=0} = 0$

[3] Where by absolute value we denote coordinate-wise

# 6 Dependent Single Time-Series Samples

Thus far we have assumed that we are working with $n$ independent time series, each of duration $m$. Though this is applicable to many settings where we have panel data with many units over time, in some other settings it is unreasonable to assume that we have many units over time, but rather that we have the same unit over a long period. In this case, we would want to do asymptotics as the number of periods grows. Our goal is still to estimate the dynamic treatment effects (i.e. the effect of a treatment at period $t$ on an outcome in period $t + \kappa$, for $\kappa \in \{0, \ldots, m\}$) for some fixed look-ahead horizon $m$.

These quantities can allow us to evaluate the effect of counterfactual treatment policies on the discounted sum of the outcomes, i.e. $\sum_{t=0}^{\infty} \gamma^t Y_t$ for $\gamma < 1$. We can write the counterfactual value function for any non-adaptive policy as: $V(\tau) = \sum_{t=0}^{\infty} \gamma^t \sum_{q \leq t} \theta_{t-q} \tau_q$. Assuming outcomes are bounded, the effect $\sum_{q \leq t} \theta_{t-q} \tau_q$ on any period $t$ can be at most some constant. Thus taking $m$ to be roughly $\log_\gamma(n)$, suffices to achieve a good approximation of the effect function $V(\pi)$, since the rewards vanish after that many periods, i.e. if we let: $V_m(\tau) = \sum_{t=0}^{m} \gamma^t \sum_{q \leq t} \theta_{t-q} \tau_q$, then observe that: $\|V_m(\tau) - V(\tau)\| \leq O(\gamma^m)$. Thus after $m = \log_{1/\gamma}(n)$, we have that the approximation error is smaller than $1/\sqrt{n}$. Thus it suffices to learn the dynamic treatment effect parameters for a small number of steps. To account for this logarithmic growth, we will make the dependence on $m$ explicit in our theorems below.

For any $m$, we will estimate these parameters by splitting the time-series into sequential $B = n/m$ blocks of size $m$. Then we will treat each of these blocks roughly as independent observations and we denote the resulting estimate as $\hat{\theta}$. The main challenge in our proofs is dealing with the fact that these blocks are not independent but serially correlated. However, we can still apply techniques, such as martingale Bernstein concentration inequalities and martingale Central Limit Theorems to achieve the desired estimation rates.

The other important change that we need to make is in the way that we fit our nuisance estimates. To avoid using future samples to train models that will be used in prior samples (which would ruin the martingale structure), we instead propose a progressive nuisance estimation fitting approach, where at every period, all prior blocks are used to train the nuisance models and then they are evaluated on the next block. We present the normality theorem and defer the finite sample MSE result to the appendix.

**Theorem 5.** *Let $\mathcal{F}_b$ denote the filtration up until block $b$. Suppose that $\mathbb{E}[\zeta_t \zeta_t' \mid \mathcal{F}_b] \succeq 2\lambda I$ and that each coordinate $\hat{h}$ of each nuisance model $\{q_\kappa, p_{\tau,\kappa}\}_{\tau,\kappa}$ satisfies: $\forall b \geq B/2$*

$$\mathbb{E}_{\hat{h}}[\|\hat{h} - h^*\|_{b,2}^4]^{(1/4)} \leq o(B^{-1/4})$$

*Moreover, suppose that:* $\left\| \sum_{\tau=0}^{m} |\mathbb{E}\left[ \tilde{T}^*_{m-\tau,m-\kappa} \zeta_t' \mid \mathcal{F}_b \right]| \right\|_\infty \leq c_m$ *and that $m$ satisfies that $mC_m = o(\sqrt{B})$, where $C_m := \max\left\{ 1, \left( \frac{c_m}{2\lambda - O(m\,d\,\epsilon_{n,\delta})} \right)^{m+1} \right\} m$. Then:*

$$\sqrt{B}\Sigma^{-1/2}M(\hat{\theta} - \theta_0) \to N(0, I)$$

*where $M, \Sigma$ are the same as in Theorem 4.*

# 7 Generalization to Structural Nested Mean Models (SNMMs)

We now generalize our results by drawing a connection between our algorithm and $g$-estimation of structural nested models in biostatistics [27] and show how our DynamicDML algorithm can be applied almost verbatim with a small change of variables and estimate dynamic effects in a much more general setting. Consider an arbitrary time-series process $\{X_t, T_t\}_{t=1}^{m}$, with $X_t \in \mathcal{X}_t$ and $T_t \in \mathcal{T}_t$. Let $Y$ denote some final outcome of interest. For any time $t$, let $\bar{X}_t = \{X_1, \ldots, X_t\}$ and $\bar{T}_t = \{T_1, \ldots, T_t\}$, denote the sequence of the variables up until time $t$ and similarly, let $\underline{X}_t = \{X_t, \ldots, X_m\}$ and $\underline{T}_t = \{T_t, \ldots, T_m\}$. We will also denote with $\bar{x}_t, \bar{\tau}_t, \underline{x}_t, \underline{\tau}_t$, corresponding realizations of the latter random sequences. Let $\pi = (\pi_1, \ldots, \pi_m)$ denote any dynamic policy, such that for each $t$, $\pi_t$ maps a history $\bar{x}_t, \bar{\tau}_{t-1}$ into a next period action $\tau_t$. For any such dynamic policy, let $Y^{(\pi)}$ denote the counterfactual outcome under policy $d$. For any static policy $\tau \in \times_{t=1}^{m} \mathcal{T}_t$, we will overload notation and let $Y^{(\tau)}$ denote the counterfactual outcome under this static treatment policy.

Moreover, for any two policies (static or dynamic) we will be denoting with $(\bar{\pi}_t', \underline{\pi}_{t+1})$, the policy that follows $\pi'$ up until time $t$ and then continues with policy $\pi$. We let $0 \in \mathcal{T}_t$ denote a baseline policy value, which could be appropriately instantiated based on the context.

We assume that the data generating process satisfies the *sequential conditional exogeneity condition*:

$$\{Y^{(\tau)}, \tau \in \times_{t=1}^m \mathcal{T}_t\} \perp\!\!\!\perp T_t \mid \bar{T}_{t-1}, \bar{X}_t$$

Identification of mean counterfactual outcomes $\mathbb{E}[Y^{(\pi)}]$ for a target policy of interest $\pi$ can be expressed in terms of the following conditional expectation functions:

$$\gamma_t(\bar{x}_t, \bar{\tau}_t) = \mathbb{E}\left[Y^{(\bar{\tau}_t, \underline{\pi}_{t+1})} - Y^{(\bar{\tau}_{t-1}, 0, \underline{\pi}_{t+1})} \mid \bar{T}_t = \bar{\tau}_t, \bar{X}_t = \bar{x}_t\right]$$

which corresponds to the mean change in outcome if we go to all units which received treatment $\bar{\tau}_t$ up until time $t$ and had observed state history $\bar{x}_t$ and we remove their last treatment, while we subsequently always continue with the target policy $\pi$. These functions are known as the *blip* functions [4, 29]. Theorem 3.1 of [29] shows that via a telescoping sum argument and invoking the sequential randomization condition that:

$$\mathbb{E}[Y^{(\bar{\tau}_{t-1}, \underline{\pi}_t)} \mid \bar{X}_t, \bar{T}_t = \bar{\tau}_t] = \mathbb{E}\left[Y + \sum_{j=t}^m \rho_j(\bar{X}_j, \bar{T}_j) \mid \bar{X}_t, \bar{T}_t = \bar{\tau}_t\right]$$

where $\rho_j(\bar{X}_j, \bar{T}_j) := \gamma_j(\bar{X}_j, (\bar{T}_{j-1}, \pi(\bar{X}_j, \bar{T}_{j-1}))) - \gamma_j(\bar{X}_j, \bar{T}_j)$ and hence also: $\mathbb{E}[Y^{(\pi)}] = \mathbb{E}\left[Y + \sum_{t=1}^m \rho_t(\bar{X}_t, \bar{T}_t)\right]$. Importantly, the conditioning set contains the observed $t$ periods treatment. Intuitively, each term $\rho_j$, removes from the outcome the *blip effect* of the observed action $T_j$ and adds the *blip effect* of the target action $\pi(\bar{X}_j, \bar{T}_{j-1})$. Thus for any parameterization of the blip functions $\gamma_t(\bar{x}_t, \bar{\tau}_t; \psi_t)$, if we let $H_t(\psi) := Y + \sum_{j=t}^m \rho_j(\bar{X}_j, \bar{T}_j)$, then the true parameter vector must satisfy the moment restrictions:

$$\forall t \in [m], \forall f : \mathbb{E}[H_t(\psi) \left(f(\bar{X}_t, \bar{T}_t) - \mathbb{E}[f(\bar{X}_t, \bar{T}_t) \mid \bar{X}_t, \bar{T}_{t-1}]\right)] = 0$$

If the blip functions admit a linear parametric form:

$$\gamma_t(\bar{x}_t, \bar{\tau}_t; \psi_t) := \psi_t' \phi(\bar{x}_t, \bar{\tau}_t)$$

for some known feature vector $\phi$, satisfying $\phi(\bar{x}_t, (\bar{\tau}_{t-1}, 0)) = 0$, then we can consider the subset of the moment restrictions of the form:

$$\forall t \in [m] : \mathbb{E}[H_t(\psi) \left(\phi(\bar{X}_t, \bar{T}_t) - \mathbb{E}[\phi(\bar{X}_t, \bar{T}_t) \mid \bar{X}_t, \bar{T}_{t-1}]\right)] = 0$$

Moreover, we can also subtract from $H_t(\psi)$, its conditional expectation $\mathbb{E}[H_t(\psi) \mid \bar{X}_t, \bar{T}_{t-1}]$ while maintaining the moment condition:

$$\forall t \in [m] : \mathbb{E}[\left(H_t(\psi) - \mathbb{E}[H_t(\psi) \mid \bar{X}_t, \bar{T}_{t-1}]\right) \left(\phi(\bar{X}_t, \bar{T}_t) - \mathbb{E}[\phi(\bar{X}_t, \bar{T}_t) \mid \bar{X}_t, \bar{T}_{t-1}]\right)] = 0$$

This is the doubly robust moment condition proposed by [28, 29], where it is shown that an estimator of $\psi$ based on this moment is correct if either the estimate of $\mathbb{E}[H_t(\psi) \mid \bar{X}_t, \bar{T}_{t-1}]$ or the estimate of $\mathbb{E}[\phi(\bar{X}_t, \bar{T}_t) \mid \bar{X}_t, \bar{T}_{t-1}]$ is correct. However, estimating $\mathbb{E}[H_t(\psi) \mid \bar{X}_t, \bar{T}_{t-1}]$, itself requires seamingly knowledge of $\psi$, which is cumbersome and can be im-practical (e.g. by constructing poor preliminary estimates of $\psi$). [16] (see Technical Point 21.5) note that for a binary treatment and when the target policy is the all-zero policy, then $\mathbb{E}[H_t(\psi) \mid \bar{X}_t, \bar{T}_t]$ can be estimated by regressing the outcome of the population that received zero subsequent treatment on the history. However, such a population can be quite small in practice and can have severe co-variate imbalances compared to the overall population. Moreover, this approach only applies to the case of a binary treatment and a static target policy.

Here, we can achieve a Neyman orthogonal moment for identifying $\psi$, which is sufficient for robustness to biases stemming from machine learning models used to train the nuisance components, while avoiding the cumbersome part of estimating the nuisance $\mathbb{E}[H(\psi) \mid \bar{X}_t, \bar{T}_{t-1}]$. Moreover, our approach leads to a strongly convex loss for the parameters at each step of the recursive process, which is beneficial for finite sample guarantees and subsequently for generalizing it to linear models with parameter heterogeneity with respect to exogenous covariates.

In particular, let $Q_j := \phi(\bar{X}_j, \bar{T}_j) - \phi(\bar{X}_j, (\bar{T}_{j-1}, \pi(\bar{X}_j, \bar{T}_{j-1})))$ and for any $t \leq j$, let $\tilde{T}_{j,t} := Q_j - \mathbb{E}[Q_j \mid \bar{X}_t, \bar{T}_{t-1}]$ and $\tilde{Y}_{m,t} := Y - \mathbb{E}[Y \mid \bar{X}_t, \bar{T}_{t-1}]$. Then, note that:

$$H_t(\psi) - \mathbb{E}[H_t(\psi) \mid \bar{X}_t, \bar{T}_{t-1}] = \tilde{Y}_{m,t} - \sum_{j=t}^m \psi_j' \tilde{T}_{j,t}$$

Moreover, note that since the second term in $Q_t$ only depends on the conditioning set $\bar{X}_t, \bar{T}_{t-1}$, then $\tilde{T}_{t,t} = \phi(\bar{X}_t, \bar{T}_t) - \mathbb{E}[\phi(\bar{X}_t, \bar{T}_t) \mid \bar{X}_t, \bar{T}_{t-1}]$. Thus we conclude that the true parameters must be satisfying the moment restrictions:

$$\forall t \in [m] : \mathbb{E}\left[\left(\tilde{Y}_t - \sum_{j=t+1}^m \psi_j' \tilde{T}_{j,t} - \psi_j' \tilde{T}_{t,t}\right) \tilde{T}_{t,t}\right] = 0$$

These are exactly of the same form as the moment restrictions considered in the definition of the Dynamic DML algorithm and its analysis. Thus the results we presented so far directly extend to the estimation of structural nested mean models, with a linear parameterization of the blip functions, simply by using the different definition of the residual variables $\tilde{Y}_t$ and $\tilde{T}_{t,t}$ and letting $\psi_t = \theta_{m-t}$ in the definition of Algorithm 1 and in Theorems 3 and 4. Moreover, note that the nuisance models that are trained in the first stage have a larger conditioning set which includes all past history of states and treatments $\bar{X}_t, \bar{T}_{t-1}$. If we made further restrictions that there was a "funnel state" $S_t$ at each period that summarizes the history and such that any dependence of the future to the past is going through that funnel state, then conditioning only on that state would have been sufficient. This is what we essentially did in the linear Markovian model. Moreover, the linear Markovian model with a static policy $\tau$, is a special case where the blip functions take the simple form: $\gamma_t(\bar{x}_t, \bar{\tau}_t) = \theta_{m-t}' \tau_t$ and are target policy independent.

Thus our Dynamic DML algorithm extends to the estimation of the structural parameters in a structural nested mean model for any target dynamic policy $d$ and any user defined baseline policy $\bar{0}$ (referred to as a double regime structural nested mean model). It allows for the estimation of the nuisance functions with arbitrary machine learning algorithms, subject to a relatively slow mean-squared-error condition and reduces estimation to simple regression and classification oracles in the first stage, with only a simple linear system of equations in the second phase, which can also be solved in linear time in a recursive manner.

**Dynamic effect heterogeneity** Finally, we note that our moment condition that identifies each parameter $\psi_t$ is the derivative of a square loss and can be written as the solution to the following square loss minimization problem: $\min_{\psi_t} \mathbb{E}\left[\left(\tilde{Y}_t - \sum_{j=t+1}^m \psi_j' \tilde{T}_{j,t} - \psi_t' \tilde{T}_{t,t}\right)^2\right]$, fixing the solution $\psi_j$ for any $j > t$, from previous iterations. This allows us to generalize the Dynamic DML to the case where we allow non-parametric heterogeneity in the parameters $\psi_t$, with respect to an exogenous fixed covariate vector of each sample, denoted as $X_0$, i.e. $\gamma_t(\bar{x}_t, \bar{\tau}_t) = \psi_t(x_0)' \phi(\bar{x}_t, \bar{\tau}_t)$, for a known feature map $\phi$ and unknown heterogeneous parameters $\psi$. Thus we can essentially generalize the $g$-estimation approach to SNMMs to allow for infinite dimensional parameters of the blip functions, as long as the input to these infinite dimensional parameters is fixed and not changing endogenously by the treatments (e.g. fixed characteristics of a unit). This can be achieved by simply minimizing recursively the square loss:

$$\min_{\psi_t(\cdot)} \mathbb{E}\left[\left(\tilde{Y}_t - \sum_{j=t+1}^m \psi_j(X_0)' \tilde{T}_{j,t} - \psi_t(X_0)' \tilde{T}_{t,t}\right)^2\right]$$

over arbitrary function spaces $\psi_t(\cdot)$. Using techniques from the recently introduced orthogonal statistical learning framework [12], we show in the appendix that this estimation method provides mean-squared-error guarantees on the recovered heterogeneous parameters $\psi_t$, that are robust to errors in the nuisance functions. This heterogeneous extension can also be viewed as an analogue of the RLearner meta-learner algorithm [24], generalized to the dynamic treatment regime setting.

# 8 Experimental Results

We consider data drawn from the DGP presented in Equation (1), with a linear observational policy: $p(T_{t-1}, X_t) := C\,T_{t-1} + D \cdot X_t$, with $X_0, T_0 = 0$ and $\epsilon_t, \zeta_t, \eta_t$ standard normal r.v.'s. (recall that $d$ is the number of treatments and $p$ the number of state variables). We consider the instance where: $A_{ij} = .5$, for all $i \in [p]$, $j \in [d]$, $B = .5\,I_p$, $C = .2\,I_d$, $D[:, 1:2] = .4$, $D[:, 3:p] = 0$, $\mu[1:2] = .8$. We consider settings where the effect is constant, i.e. $\theta_0 \in \mathbb{R}^d$ or heterogeneous, where $\theta_0(X) = \theta_0 + \langle \beta_0, X[S] \rangle$ for some known low dimensional subset $S$ of the states. We compare the dynamic DML to several benchmarks. The results are presented in Figures 1, comparing the estimates of the dynamic DML algorithm to a number of other alternatives on a single instance.

They fall into two categories. In the "static" set of approaches, each of the contemporaneous and lag effects is estimated one at a time, either by direct regression or (static) DML. So for example, to estimate the one period lag effect $\theta_1$, we would regress $Y_t$ on $T_{t-1}$, with controls $X$. We consider direct regression with no controls ("no-ctrls"), direct and DML with controls from the inital period (i.e $X_{t-1}$, "init-ctrls" and "init-ctrls-dml") and direct and DML with controls from the same period as the outcome (i.e $X_t$, "fin-ctrls" and "fin-ctrls-dml"). As an alternative to all of these, we try a "direct" dynamic approach, where initially we estimate $\theta_0$ using a lasso regression of $Y_t$ with all the controls and past treatments, and return the coefficient on $T_t$, and then "peel" off the estimated effect as in the main text before running another Lasso regression of $Y_t - \theta_0 T_t$ on $T_{t-1}$ to get the first lag effect etc. So this approach incorporates the peeling effect but doesn't do any orthogonalization. The point estimate for $\theta_0$, $\theta_1$ and $\theta_2$ are depicted in the three panels of Figure 1 and the error bars correspond to the constructed confidence interval. For all three, the dynamic DML is relatively close to the truth and the confidence interval contains the truth. The remaining approaches are not, although for the contemporaneous effect the approaches with final period controls have similar performance - it is really in the lagged effects that the differences become most apparent. Subsequently we run multiple experiments to evaluate the performance of DynamicDML. In each setting, we run 1000 Monte Carlo experiments, where each experiment draws $N = 500$ samples from the above DGP and then estimated the effects and lag effects based on our Dynamic DML algorithm. Figure 2 considers the case of two treatments, and shows that the algorithm performs well in terms of giving reasonable coverage guarantees - for a nominal 95% coverage, actual coverage varies from 91% to 94.5%. The right panel shows that the average estimates are close to the truth. In Figure 5 in Appendix E we repeat the experiments with $N = 2000$, and actual coverage is now tightly in the range 94% to 95%, and the average estimates remain relatively unbiased. Finally, in the appendix we explore a scenario with heterogeneous effects and find qualitatively similar performance.

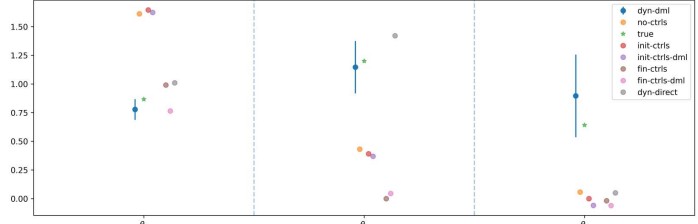

Figure 1: Comparison of DynamicDML (with confidence intervals) with benchmarks on a single instance. $n = 400$, $n_t = 2$, $n_x = 100$, $s = 10$, $\sigma(\epsilon_t) = .5$, $\sigma(\zeta_t) = .5$, $\sigma(\eta_t) = .5$, $C = 0$

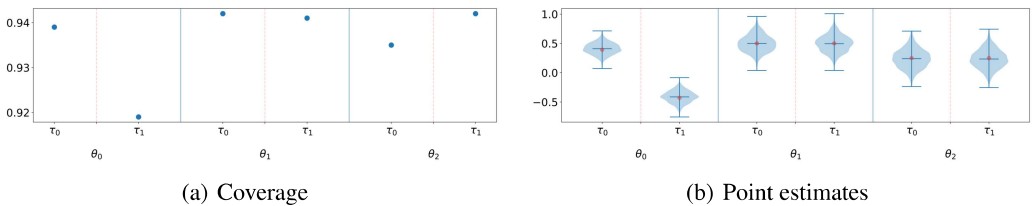

(a) Coverage            (b) Point estimates

Figure 2: $n = 500$, $n_t = 2$, $n_x = 450$, $s = 2$, $\sigma(\epsilon_t) = 1$, $\sigma(\zeta_t) = .5$, $\sigma(\eta_t) = 1$

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
