# OpenReview forum: "Double/Debiased Machine Learning for Dynamic Treatment Effects"
_NeurIPS.cc/2021/Conference — NeurIPS 2021 Poster_

### Official Review · Reviewer_bwP4 · 2021-07-02

**Rating:** 6
**Confidence:** 3

**Summary:**

The authors propose an extension of the double/debiased machine learning framework to estimate the dynamic effects of treatments and apply it to a concrete **linear Markovian high-dimensional state space model** and to general structural nested mean models. The method is based on a sequential regression peeling process, which we show can be equivalently interpreted as a Neyman orthogonal moment estimator. The authors prove both finite sample and asymptotic bounds for the estimation method. The experiments are super-minimal and do not provide convincing evidence about the algorithm's empirical performance.

**Limitations And Societal Impact:**

1. The algorithm only applies to a certain MDP. In the SNMMs, they have the classical causality assumptions such as *sequential conditional exogeneity condition*. The authors need to acknowledge these limitations and prevent a false sense of over-confidence in their results when these assumptions are violated.
2. Otherwise, the problem solved in this paper is well-studied and can potentially have a positive societal impact in reducing biases in our estimations.

**Main Review:**

## Strengths
1. Algorithm 1 is rather simple, which increases its potential impact.
2. The theoretical analyses seem correct.
3. the idea of having a Z-estimator using the moment conditions to get the desirable theoretical rates is interesting.

## Weaknesses
1. The experiments are not convincing. The authors should report the uncertainty intervals for the `dyn-direct`   algorithm too.
2. The results only hold for the MDP in Eq. (1), which is slightly less restrictive than the linear model.
3. The authors should pick a less generic title. There are other double robust dynamic treatment effect estimation algorithms.
4. The notation is rather messy. For example, the authors use $d$ for the dimension of the treatment (line 84) and dynamic policy (line 232).
5. The assumptions are rather spread out during the text, especially in Section 7.

## Further Comments
1. Please enable line numbers for the Algorithm.
2. The equation at line 115 needs more explanation. Where did the shocks $\delta_m$ come from?

**Time Spent Reviewing:**

10

---

> ### Author Response · Authors · 2021-08-10
> **Review response**
>
> We thank the reviewer for the elaborate review and comments and address each one in detail:
>
> 1. The dyn-direct algorithm does not offer confidence interval construction, since it uses a lasso algorithm to estimate the target parameters and hence one cannot simply add confidence intervals around parameters of a lasso estimator. One could provide bootstrap confidence intervals, but this would be computationally prohibitive, since around 10k bootstrap samples would be required for sufficient good approximation of uncertainty. The main advantage of Neyman orthogonal estimators, is exactly that they allow construction of asymptotically valid intervals.
> 2. As we note in Section 7 our results go well beyond the linear markovian setting, to any linear SNMM, which is a class of models that have been widely studied and applied in biostatistics.
> 3. We will add: “via g-Estimation” to the end of our title, to point to the type of estimation technique we use.
> 4. We will switch d to pi for the dynamic policy.

---

### Official Review · Reviewer_U835 · 2021-07-10

**Rating:** 6
**Confidence:** 3

**Summary:**

Contributions of this paper are the following:
- Extending Double/Debiased Machine Learning (DML) theories from the static treatment regimes to dynamic treatment regimes for considering a semi-parametric Markovian model with the flexible high-dimensional state, under the partial linear model settings and its generalization to Structural Nested Mean Model (Section 7).
- In extending, the authors proposed to employ (minimal) parametric assumptions to avoid the ill-posed problem and deal with high-dimensional covariates.

**Limitations And Societal Impact:**

Some of limitations are commented in the Main review.
- I'd like to propose to explain the possibility of practical experiments (or possible scenarios) corroborating with the proposed theories.
- What are the parametric assumptions in SNMM? Could they be explained more explicitly?
- Can Thm. 4 be more extended to cover doubly robustness results?

**Main Review:**

This paper focuses on extending DML theories to the dynamic treatment regimes by employing semi-parametric assumptions to avoid problems that could occur in nonparametric settings. This paper's contributions are significant given that partial linear setting and dynamic treatment regimes have important practical implications.

Here are comments for helping readers understand the results and some questions that readers might be curious about.
- 1. In Eq. (1), What is 'p' in p(Tt-1, Xt)? I couldn't find its definition.
- 2. The model in Eq.(1) looks somewhat restricted because it confines the relationship between variables as linear. It would be better if practical motivations or examples that could be covered by Eq. (1) are provided.
- 3. Sec. 7 look significant because these address my concern in comment 2 (above). It would be helpful if results in Sec. 7 are formalized as Lemmas or Theorems.
- 4. I'd like to see if practical experiments corroborate with the proposed theories, given that the settings discussed in this paper are simple and highly practical (partial linear model with dynamic treatment regimes)
- 5. Is it nontrivial to extend results in Thm. 4 for showing doubly robustness (robustness against model misspecification)?

I'd like to suggest highlighting results in Section 7 or even consider reorganizing to show them first. Personally, even if I liked strong theories and highly practical settings of the work at first glance, I was concerned that settings in Sec. 2 looks restricted. But, my concerns were addressed by results in Section 7. Having said that, it'd be more attractive if results in Section 7 come earlier.

**Time Spent Reviewing:**

4 hours

---

> ### Author Response · Authors · 2021-08-10
> **Review response**
>
> We thank the reviewer for the elaborate review and comments and address each one in detail:
>
> 1. The function p could be any propensity function. We will add a note right after equation 1.
> 2. We presented the linear Markovian setting first for expository purposes, but we note that our SNMM extension goes well beyond linear Markovian assumptions. Even so we do think that the fact that we allow for a high-dimensional state adds a lot of flexibility even for the linear markovian setting to be used in practice, since the user can simply consider high-dimensional feature expansions of low dimensional states to capture non-linearities.
> 3. We will move the main theorem from section 7 in the main text, as indeed it provides significant extension. Space constraints were very tight.
> 4. (see response to Reviewer 1)
> 5. Double robustness: One can actually show partial double robustness, while maintaining the qualitative properties of our solution. In the sense that one can show by a more careful analysis of the remaining terms in the proof that we require that the propensity model achieve n^{¼} rates and that the product of the propensity and regression model errors be n^{½}. Thus if the propensity model is very good, the regression model is not that important. Maintaining the property that our solution corresponds to the derivative of a strongly convex loss at each step of the peeling, which is important for many of our extensions and for better finite sample properties, and simultaneously also achieving double robustness might be infeasible.
> 6. We will re-organize and move the definitions of SNMMs first and then present that stylized setting and then the SNMM main result, so that the reader has a map forward to the more general results.
> 7. Parametric assumptions in SNMM: one potential illucidating explanation of the implicit assumptions of linear SNMMs, as is noted in Chakraborty and Moodie (ref[4] in the paper) is that linear SNMMs make in some sense equivalent modelling assumptions as what is being made in Q-learning with linear Q functions, a quite standard approach in reinforcement learning (though the estimation algorithm of g-estimation is different than linear Q-learning). We will add such a discussion.

---

### Official Review · Reviewer_FZDr · 2021-07-16

**Rating:** 7
**Confidence:** 1

**Summary:**

The paper proposes a DML (double machine learning) based estimation methodology of dynamic treatment effects. Writing the conditional moment restriction on the "adjusted" outcome allows the authors to bring the methodology of orthogonal score to dynamic treatment effects. Using this insight, on a partially linear model, the authors propose a progressive (called peeling) algorithm where  the "adjusted" outcome can be regressed on the treatment and covariates to estimate the coefficient of the treatment. The authors then show that under slower than root-n estimation rates of the nuisance functions, the theta parameters still enjoy root-n consistent estimates. The authors then discuss using a single sequence of observations to estimate the effect of an intervention (dynamic) on a discounted outcome. The method is then shown to generalize to non-linear sequential models (SNMMs) which a much more general class than partially linear models.

**Limitations And Societal Impact:**

The paper solves the propose problem well enough but it must be noted that structural nested mean models do not contain all possible models. But I believe this is not a significant limitation.

**Main Review:**

The paper is mostly clearly written despite the inevitable denseness of equations when dealing with effect estimation in sequential data. With the generalization of the method to nested mean models, a lot of my concerns about generality are alleviated but it is unclear how much the knowledge of the representation plays a role in estimation. For example, it seems like while the method does not restrict the class of representation phi(X), it cannot be learned as  part of the process. Could the authors comment on this?

**Time Spent Reviewing:**

4

---

> ### Author Response · Authors · 2021-08-10
> **Review response**
>
> We thank the reviewer for the elaborate review and comments and address each one in detail:
>
> 1. Indeed for parametric rates and for root-n normality, one needs to make a parametric assumption on some part of the data generating process and here we do so via the blip functions (in the SNMM section). For the main results, these representations phi(X) need to be hard-coded. However as we show in the appendix sections, we can greatly relax the literature on SNMM and learn partly the form of phi(X). 1) we can allow for high-dimensional phi(X) (appendix A.2), and apply a recursive lasso algorithm, which essentially will learn the relevant parts of the representation. Thus this allows learning of the feature map among a high-dimensional candidate set, under sparsity assumptions. 2) we can allow for arbitrary heterogeneity of the parameters in the linear representation (appendix A.1), i.e. <theta(X0), phi(X_t)>, with respect to non-time varying characteristics of the treated unit. This leads to a much more flexible model selection approach, on how blip function change with fixed treatment characteristics.

---

### Official Review · Reviewer_EKsS · 2021-08-03

**Rating:** 6
**Confidence:** 3

**Summary:**

This paper combines the $g$-estimation approach to identifying dynamic treatment effects in structural nested mean models with orthogonal ML to obtain asymptotically normal estimates of the treatment effects. The main benefits of using orthogonal ML to estimate the treatment effects is it allows high-dimensional state spaces and is statistically efficient (with cross-fitting).

**Limitations And Societal Impact:**

The authors discussed limitations of their method. There are not direct potential negative societal impacts of their work.

**Main Review:**

The most interesting part of this paper is the identification strategy (Lem 1 and Thm 2). Although both seem to follow from straightforward manipulations of (1), I suggest the authors elaborate on the identification strategy. In particular, I suggest the authors include the proofs of Lem 1 and Thm 2 in the main paper. This better motivates the "peeling" method. In light of the page limit, I suggest the authors defer either the simulation results or one of the extensions (sec 6 or 7) to appendices.

Another way that the paper can be improved is by demonstrating the applicability of the proposed method on real data. The linear MDP imposes strong restrictions on the data generating process, thereby limiting the applicability of the method. I suggest the authors show that the method works on real data to assuage concerns regarding the proposed method's applicability.

**Time Spent Reviewing:**

1.5

---

> ### Author Response · Authors · 2021-08-10
> **Review response**
>
> We thank the reviewer for the elaborate review and comments and address each one in detail:
>
> 1. We will include one of the two proofs in the camera ready given the extra space.
> 2. We note that in the main extension section on SNMM the method extends well beyond the linear MDP setup and hence its applicability is much wider than the expository linear MDP example. It was not clear what conclusion to draw on the quality of the method from real world experiments, given that the ground truth is not known, which is one reason we opted to measure performance in synthetic data. We have already applied our method on real-world datasets and we can add semi-synthetic experiments that use some qualitative characteristics of the real world data. Moreover, linear SNMMs have already been heavily used in biostatistics and hence we expect that our methodology will be applicable to any setting where linear SNMMs have been already applied. However, many of the clinical trial datasets used in SNMM papers that we could identify were not publicly accessible.

---

### Comment · Area_Chair_eqKh · 2021-08-19
**Relationship**

Dear authors,

Could you expand on the relationship of the submission with [9, 8, 39, 12]? The end of section 1 mentions a "vague" relationship and more precision would be helpful. I am particularly interested in comparisons with the results in these works that also provide valid confidence intervals, and what are sets of assumptions (if any) would make it possible to compare the results in those works to the theorems of the submission.

Best wishes,
Area Chair

---

> ### Author Response · Authors · 2021-08-19
> **Relation to [9,8,39,12]**
>
> Here is a more elaborate description of the connection. Potentially the closest to our work is that of Deshpande et al 2019 (the relationship and comparison to the rest of the papers in this line of work is similar in spirit so we focus on Deshpande et al to highlight concrete differences). The work of Deshpande et al 2019 deals with the following setting: we observe data y_t, X_t, where for each t, the co-variate X_t that we observe is adaptively chosen, based on past observations (e.g. of past y_t and X_t). Moreover X_t is high dimensional and we assume a linear model y = a*X_t + epsilon_t. The goal of that work is to perform inference on some coefficient a_i, even when data are adaptively collected. Here are the differences to our work:
>
> 1) The best way to connect to our setting is to make T_{t} be the i-th coordinate of X_t and then the work of Deshpande et al can be viewed as performing inference on the treatment effect of T_t on y_t, while controlling for the rest of the X_t. The adaptive collection of that work matches the fact that treatment is chosen in by an adaptive dynamic policy in our work and that covariates are also auto-regressive.
>
> 2) However, the parameter a_i  is solely the “contemporaneous” or “Same-period” effect of the treatment. The work of deshpande et al provides no method for inference on the dynamic treatment effect, which is the effect of treatment at period t on few steps ahead. To achieve this a very different identification argument is needed and the target dynamic effect parameter cannot be thought as the result of a simple regression (square loss minimizer) of y on X, which is a crucial aspect of that work. You need a repeated peeling process to identify this effect, which is the main result of our identification sections. Thus the target parameter on which we perform inference is qualitatively very different than that in Deshpande et al (i.e. dynamic effect vs contemporaneous effect) and is not captured by all this prior work (neither other papers in the list).
>
> 3) In the case were we only care about inference on the contemporaneous effect, then indeed our work is closely related to that line of work. In particular, the closest related section is that on “estimation from a single correlated chain”. This is the only section where we look at the case of estimating form a single chain of serially correlated data. The rest of the paper, is working in vastly different setup where we assume access to n samples of m-length chains. This is the typical setting in the work on the dynamic treatment regime. In this setting of n independent samples of m-length chains, the techniques in Deshande et al and in the rest of the papers, are in some sense irrelevant. The reason is that in this setting, the asymptotics are performed as n grows and m is constant. In that case, we do not have to deal with the fact that the covariance matrix of a single serially correlated chain does not converge and hence asymptotic normality arguments become very complex and require more fancy adaptive sample re-weighting. These techniques are not required when observing n independent chains.
>
> 4) In the case of a single observed chain, and if we only care about contemporaneous effects, then our setting does collapse to the settings addressed by this line of work and our Theorem 5 is an alternative inference result, based on Neyman orthogonal moment equations, as compared to the work of Deshpande  et al, which uses a debiased lasso style argument. However, contemporaneous effects is not the focus of this work and hence, for this reason we think this work is tangentially related. Moreover, our results on a single long chain, estimate the more complex dynamic effects and not just contemporaneous effects. However, we note that for that result (Theorem 5) we do need to assume the simplified linear Markovian setting and that the noise variables are iid (this is the reason why there is no analogue of Thoerem 5 for the more general SNMM section; where we only provide results for n iid, m-length chains). This allows us to prove inference by invoking Martingale CLT theorems without adaptive sample-reweighting.
>
> 5) It is a very interesting avenue for future research if our results on inference for dynamic effects, for a single dependent time series, without the iid noise assumption (especially the fact that zeta_t; the noise term in the treatment policy; is iid is harsh and typically wont hold for binary treatments). This extension would need to combine identification and estimation strategy results from our work, with sample re-weighting from the line of work of (Deshpande et al., 2018, 2019; Zhang et al., 2020; Zhan et al., 2021b; Hadad et al., 2021; Bibaut et al., 2021; Zhan et al., 2021a) to get at an estimation strategy where the jacobian of the estimating moment equation will converge asymptotically and hence we can apply some form of Martingale CLT. We view this as an orthogonal extension to our main contribution, which is the Neyman orthogonal estimation of dynamic effects. Albeit a very interesting extension. Moreover, the single dependent time series setting is a secondary contribution of our paper and for this reason we did not press further on this extension.
>
> 6) Our work extends well beyond the linear Markovian setting to the SNMM setting and as we show in the appendix, even to heterogeneous effects and high-dimensional target parameters. These aspects are not captured at all by the line of work mentioned.
>
> In sum, we think that our work offers an orthogonal contribution in the realm of inference from adaptively collected, closer related to the literature on the dynamic treatment regime and less on the more recent literature on inference from adaptively collected data. Albeit as we describe above there is a concrete meeting point, which could be a fruitful avenue for future work and extensions of our work.
>
> We will definitely make sure to elaborate on this, definitely in the appendix of the camera ready and expand a bit more in the intro paragraph. We could not expand further in the submission due to space constraints and given the subtleties of the relations as described above.

---

### Decision · Program_Chairs · 2021-09-27

**Decision:**

Accept (Poster)

**Comment:**

There is a consensus among reviewers that the submission provides a solid
contribution for the challenging problem of estimating dynamic treatment
effects, i.e., estimating at time t+m the the causal effect of a treatment
given at time t. Although based on known DML/Neyman orthogonal scores concepts,
the methodology leverages a novel peeling argument which may have further applications in sequential causal inference problems.
Finally no major issues with the theory were raised in the reviews.
For all these reasons, I recommend to accept the paper.

The authors are encouraged to include in the camera-ready version of the paper
a semi-synthetic experiment that use characteristics from real world datasets,
and to implement the improvements that emerged during the discussion (e.g.,
avoiding d for the policy, expanding on the relationship with [8, 9, 39, 12] in
the related works section)
I also encourage the authors to discuss briefly
the practicability of verifying the various rate conditions appearing
in the assumptions, e.g., when is it possible to observe from the data
whether these rate conditions are satisfied for a given dataset.